# Transcriptome and Proteome Co-Profiling Offers an Understanding of Pre-Harvest Sprouting (PHS) Molecular Mechanisms in Wheat (*Triticum aestivum*)

**DOI:** 10.3390/plants11212807

**Published:** 2022-10-22

**Authors:** Sang Yong Park, Woo Joo Jung, Geul Bang, Heeyoun Hwang, Jae Yoon Kim

**Affiliations:** 1Department of Plant Resources, College of Industrial Science, Kongju National University, Yesan 32439, Korea; 2Institute of Life Science and Natural Resources, Korea University, Seoul 02841, Korea; 3Research Center for Bioconvergence Analysis, Korea Basic Science Institute, Cheongju 28119, Korea

**Keywords:** wheat (*Triticum aestivum*), abiotic stress, pre-harvest sprouting (PHS), RNA-seq, transcriptome, proteome analysis, DEGs, DEPs, functional annotation, Gene Ontology, seed germination, seed dormancy, metabolite mechanisms

## Abstract

While wheat (*Triticum aestivum* L.) is a widely grown and enjoyed crop, the diverse and complex global situation and climate are exacerbating the instability of its supply. In particular, pre-harvest sprouting (PHS) is one of the major abiotic stresses that frequently occurs due to irregular climate conditions, causing serious damage to wheat and its quality. In this study, transcriptomic analysis with RNA-seq and proteomic analysis with LC-MS/MS were performed in PHS-treated spikes from two wheat cultivars presenting PHS sensitivity and tolerance, respectively. A total of 13,154 differentially expressed genes (DEGs) and 706 differentially expressed proteins (DEPs) were identified in four comparison groups between the susceptible/tolerant cultivars. Gene function and correlation analysis were performed to determine the co-profiled genes and proteins affected by PHS treatment. In the functional annotation of each comparative group, similar functions were confirmed in each cultivar under PHS treatment; however, in Keumgang PHS+7 (K7) vs. Woori PHS+7 (W7), functional annotations presented clear differences in the ”spliceosome” and ”proteasome” pathways. In addition, our results indicate that alternative splicing and ubiquitin–proteasome support the regulation of germination and seed dormancy. This study provides an advanced understanding of the functions involved in transcription and translation related to PHS mechanisms, thus enabling specific proposals for the further analysis of germination and seed dormancy mechanisms and pathways in wheat.

## 1. Introduction

Wheat (*Triticum aestivum* L.) is one of the most widely produced major food crops, being the most-cultivated species among the *Triticum* genus, and it is typically called common wheat or bread wheat [1,2,3,4,5]. Wheat has various genetic characteristics, such as hardness, autumn/spring wheat, spikes, and seed coat color, which are considered as important factors in the processed food product [6,7]. Because it has high value as a major food, wheat also satisfies personal preferences through various functional factors [8,9,10]. As wheat usually requires a milling step, the grain quality in the harvest season is considered to be a major factor in the end-use quality of wheat. The current global climatic conditions cause a variety of abiotic stresses. In particular, biotic and abiotic stress damage can lead to a drastic reduction in wheat grain quality as a food, as well as wheat yield [11,12,13]. Excess moisture, as an abiotic stress, causes increased damage to wheat grains. PHS occurs frequently under excessive moisture conditions, causing great damage to wheat, in terms of grain quality [14,15].

Pre-harvesting sprouting (PHS) is a phenomenon in which germination inhibitors are dissolved in the kernels or spikes under frequent rainfall and excessive moisture conditions, breaking seed dormancy and resulting in germination before harvest [16,17,18,19]. PHS is a major problem that severely decreases the quality of wheat seeds, and this representative abiotic stress has caused critical damage worldwide, notably in East Asia [20,21,22]. In the Korean wheat cultivation environment, a drastic increase in temperature in the rainy season during the harvest period leads to PHS. Therefore, understanding the PHS mechanism during the grain development stage is one of the most important research aspects in the wheat breeding field [23]. Additionally, a physiological and genetic understanding of seed development, germination, and dormancy is necessary to address the problem of moisture stress.

Plant genome studies remain challenging, due to their genome size, ploidy, and frequent genome duplication [24]. The genome of *Arabidopsis*, a widely used model plant, is diploid and small, at only 125 Mb [25]. However, the genome of common wheat has three separate sub-genomes (ABDs) with a large number of repeating elements (hexaploids), such that whole-genome sequencing still has not been completed [26,27]. Transcriptome or partial genome sequencing have recently been considered as alternative analysis methods for wheat genome research. RNA-seq can be helpful in the development of breeding materials, as it allows for the selection, not only of the already known transcriptome, but also of crops in which genome sequencing has not been completely interpreted, such as wheat [28,29]. Additionally, RNA-seq enables the identification of candidate genes and expression analysis through quantified mRNA sequencing, along with microarray technology in RNA analysis research [30]. Moreover, gene of interest (GOI) searches and gene expression analysis are being actively used in the study of abiotic stresses, such as drought, heat, salinity, cold damage, and moisture stress [31,32,33,34,35]. The identification of DEGs (differentially expressed genes) though RNA-seq under different stress conditions may provide evidence for an understanding of the genetic effect and metabolic mechanism of wheat. However, RNA-seq data still require validation, as the transcriptome provides a dynamic range of data, according to the plant conditions [36,37,38].

Proteins play a major role in regulating almost all cellular processes. The proteome creates a highly diverse biological network and enables the normal operations of the phenotype, cellular morphology, and function of plant organisms [39]. Recent proteomic technologies mainly aim to measure proteins, both qualitatively and quantitatively [40,41]. Quantitative proteomic analysis has been conducted in an attempt to screen proteins between GM soybeans and non-GM soybeans, and the quantitative proteomic analysis of DEPs between potato cultivars treated for disease infection has also been performed [42,43]. On the other hand, qualitative proteomic studies on crotonylation allow for the determination of attributes that are specifically related to plant growth and development, and as such, they are being actively discussed in various crops [44,45,46]. Advances in chromatography-coupled mass spectrometry (MS) are closing the gap in proteomic plant research. MS analysis is another essential element for the validation of gene expression, as it confirms the actual presence of proteins [47]. Additionally, utilizing MS/MS spectra for predictable protein sequences, sequence variants, or indeed, whole-genome translations can further improve gene annotation or identify genes that have been missed in transcriptome analysis or in silico annotation [48]. Therefore, proteogenomics provides value, which is helpful in the analysis of plant gene annotation, and it can be expected that the utilization of proteogenomics in plant research will expand [48,49]. Compared to humans and animals, plants have only recently begun to be the subject of active proteomic research [50]. Several abiotic stress studies based on proteomic research have been reported in plants. In particular, a study on the improvement of drought resistance through the inhibition of enzymes related to the biosynthesis of lignin, flavonoids, and fatty acids caused by drought stress in tea trees has been conducted [51]. A comparative analysis of protein expression between maize cultivars identified that the accumulation of antioxidant enzymes could be affected by increased drought tolerance, and it was confirmed that the difference in drought resistance was in accordance with the lignin content of roots [52]. An increase in S-adenosylmethionine synthetase 2 (*SAMS2*) confirmed that the inhibition of ethylene production due to polyamine accumulation delayed leaf aging [53]. Intracellular protein changes caused by gradual drought stress has recently been investigated in a model plant, *Arabidopsis thaliana* (L.) Heynh. [54]. Differentially expressed protein (DEPs) analysis and metabolism studies under drought, salinity, and water stress conditions have been performed to assess wheat seed protein expression [55]. Furthermore, complementary studies of the transcriptome associated with the proteome are considered valuable studies with advanced reliability and reproducibility for the determination of plant responses under stress conditions [56].

In this study, molecular mechanisms under PHS were analyzed in two Korean cultivars, ”Keumgang” (common wheat, a PHS-sensitive cultivar) and ”Woori” (common wheat, a PHS-tolerant cultivar). RNA-seq analysis was performed to identify the DEGs associated with the PHS response and tolerance mechanism. Proteomic analysis using LC-MS/MS analysis was also performed in Keumgang and Woori independently, in order to identify DEPs at the protein level. Gene Ontology analyses were carried out, followed by a comparative analysis on the function and expression of genes in RNA and protein. Through the approach described in this study, we attempted to identify key gene annotations and understand the pathways affecting PHS and seed dormancy. This study provides the possibility of complementary and elucidated functional analyses through proteomic research data, along with the existing transcriptome research. Additionally, the expression mechanisms and pathways under PHS-induced conditions are expected to enable the causal analyses of PHS stress and a further understanding of the mechanisms underlying germination and seed dormancy.

## 2. Results

### 2.1. PHS Treatment and Germination Phenotype Analysis

A PHS induction experiment was performed in the PHS-sensitive cultivar Keumgang and the PHS-tolerant cultivar Woori. The experiment focused on implementing an environment where PHS frequently occurs in the field. The growth temperature was maintained at 28 °C/15 °C, and moisture was supplied for 12 h, such that the relative humidity was 90%. In the germination rate analysis, 171 germinated from a total of 202 Keumgang seeds, showing high PHS sensitivity (at 84.65%). On the other hand, in Woori, only 6 of 206 seeds germinated, resulting in a germination rate of 2.91%, confirming that its PHS tolerance was significantly higher than that of Keumgang (Figure 1). The results of this experiment corresponded to the results of various PHS experiments performed previously [57,58,59].

### 2.2. Transcriptome and Differential Expression Gene (DEG) Analysis

Transcriptome analysis was performed with Keumgang (K0) and Woori (W0) samples, before PHS treatment, and Keumgang (K7) and Woori (W7) samples, 7 days after PHS treatment. At first, reads for each sample were mapped to the reference genome using Tophat (v2.0.13). The total number of mapped reads in the four samples was 69,035,752 reads. Among them, 57,708,213 reads were mapped, showing 83.56% mapping coverage (Appendix A). Cuffdiff v2.2.0 was used to perform a normalization of the sorted mapping counts, followed by DEG analysis. Additionally, scatter and volcano plot analyses were carried out between comparative samples, in order to view the overall changes and gene expression (Figure 2A–D). DEGs with statistically significant differences were compressed using the cut-off under the condition of 2-fold change and a *p*-value <0.005. The four groups, K0 vs. K7, W0 vs. W7, K0 vs. W0, and K7 vs. W7, revealed 5550, 2324, 386, and 4864 compressed DEGs, respectively (Appendix A). Heatmaps were used to analyze the gene expression patterns of each comparison group for the filtered DEGs. The DEG expression profiling showed that there were 3645/1905, 1373/951, 186/200, and 1561/3303 up-/downregulated DEGs, respectively, for the K0 vs. K7, W0 vs. W7, K0 vs. W0, and K7 vs. W7 groups, respectively (Figure 2E–H).

### 2.3. GO and KEGG Enrichment Analysis of Four Comparison Groups 

Gene Ontology (GO) and KEGG analyses were carried out in order to obtain a functional understanding of the DEGs that were altered and expressed under PHS treatment. Gene Ontology is considered to be effective when one or more genes are detected in functional annotation, and the functional classification is divided into three categories: biological processes (BP), cellular components (CC), and molecular functions (MF). GO annotation profiling was performed for all DEGs, and only in cases corresponding to the four comparison groups of interest. In terms of the up-/downregulated GO terms, there were, respectively, 57,664 and 27,072 DEGs in the K0 vs. K7 group, 26,597 and 15,366 DEGs in the W0 vs. W7 group, 4,426 and 4,501 DEGs in the K0 vs. W0 group, and 25,810 and 45,386 DEGs in the K7 vs. W7 group. Afterward, GO terms were cut-off with a *p*-value < 0.005 for statistical validity. As a result, the number of up- and downregulated GO terms in the four comparison groups were 17,806, 12,291, 797, and 19,429 DEGs, respectively. Finally, the 10 most annotated up- and downregulated functional annotations for each comparison group were considered. All GO terms annotated for each group are listed in Appendix A. In the K0 vs. K7 group, germination-associated “carbohydrate metabolic process”, “hydrogen peroxide catabolic process”, “response to oxidative stress”, and the “glutathione metabolic process” was confirmed as an upregulation comment. On the other hand, “ubiquitin-dependent protein catabolic process”, “mRNA splicing via spliceosome”, “protein ubiquitination”, “SCF—dependent proteasomal ubiquitin”, and “SCF ubiquitin ligase complex” were noticeably downregulated. In the W0 vs. W7 group, the upregulated GO terms were commonly annotated with common functions, where the GO terms included the regulation of transcription, “DNA-templated”, “translation”, “developmental process”, “DNA repair”, and “ATP binding”. On the other hand, several downregulated functions were interesting in contrast to K0 vs. K7. “Glutathione metabolism”, “lipid metabolism”, and “response to water deprivation” (25 DEGs) were annotated. In particular, germination-related functions, such as “oxidoreductase activity”, “monooxygenase activity”, and “UDP-glycosyltransferase activity” were downregulated (Figure 3B). The K0 vs. W0 group had smaller scales of DEGs for functional annotation analysis than K0 vs. K7, W0 vs. W7, and K7 vs. W7. Nevertheless, the upregulated GO terms showed the highest number of annotations in the “developmental process”, “DNA-templated transcription termination”, and “chloroplast organization”. Meanwhile, in the K0 vs. W0 group, the annotated GO terms for the downregulated DEGs included “defense response”, “carbohydrate metabolic process”, “response to hydrogen peroxide”, “glutathione metabolic process”, “oxidoreductase activity”, “serine-type endopeptidase inhibitor activity”, and “glutathione transferase activity”. Some germination-related functions were confirmed to be upregulated in the K0 vs. W0 group. However, in many of the downregulated GO terms, germination-related GO terms also ranked high in the annotation (Figure 3C). Interestingly, the K7 vs. the upregulated GO term in the W7 group showed the opposite result compared to the K0 vs. K7 group. In the K0 vs. K7 group, “mRNA splicing via spliceosome”, “SCF-dependent proteasomal ubiquitin-dependent protein catabolic process”, “SCF ubiquitin ligase complex”, and “ubiquitin protein ligase activity” were the main functional annotations. The downregulated GO terms included “fatty acid biosynthesis process”, “carbohydrate metabolism process” and “NAD binding”, which were mainly involved in the germination system. “NAD binding” plays a pivotal role in the metabolic pathway of the TCA cycle (Figure 3D). The linkage GO terms identified via PHS treatment in the two cultivars are considered meaningful data for researching linkage candidate genes.

KEGG pathway analysis showed that many genes involved in multiple molecular pathways were up- or downregulated after PHS treatment (Figure 4). Although activated or inhibited pathways in Keumgang and Woori mostly showed similar tendencies, there were also pathways with distinct differences. In the K0 vs. K7 group, “Benzoxazinoid biosynthesis”, “Photosynthesis-antenna protein”, “Phenylpropanoid biosynthesis”, “Phenylalanine metabolism”, “Carbon metabolism”, and “Starch and sucrose metabolism” were upregulated, whereas “Autophagy”, “Proteasome”, “Spliceosome”, and “MAPK signaling pathway” were downregulated (Figure 4A). In the W0 vs. W7 group, “Phenylalanine metabolism”, “Phenylpropanoid biosynthesis”, and “Benzoxazinoid biosynthesis” pathways were identified as activated annotations, similar to those in the K0 vs. K7 group. Furthermore, in the suppressed pathway, “Autophagy” and “MAPK signaling pathway” showed similar expression in the K0 vs. K7 group. On the other hand, a notable result in the W0 vs. W7 group was that “TCA cycle” was significantly inhibited (Figure 4B). Although the pathway was not commented out between K0 and W0, a remarkable difference was observed between K7 and W7. In K7, “Photosynthesis”, “Phenylpropanoid biosynthesis”, and “Carbon fixation in photosynthetic organisms” showed higher expression levels than in W7. On the other hand, upregulated pathways in K7 vs. W7 showing higher expression were identified as “Spliceosome”, “Folate biosynthesis”, “Proteasome”, and “Ubiquitin-mediated proteolysis”. Especially, in the “Spliceosome” and “Proteasome” pathways, more than 70% of the genes in each pathway were upregulated in W7 (Figure 4C). From the proteomics data, we discovered 42 alternative splicing variants identified by their corresponding unique peptides, which are significantly increased in the W7 and K7, with 87 and 45, respectively (Appendix A). In order to observe the alternative splicing pattern of PHS-related genes, the Integrative Genomics Viewer was utilized according to Thorvaldsdóttir et al. [60] (Appendix A). It was shown that *MFT* (*Mother of FT and TFL1*)-*3B-1* and *MFT-3B-2* had different alternative splicing patterns between Keumgang and Woori. Although no differences were observed between K0 and W0, the second and the third exon of K7 and the fourth exon of W7 were not present in *MFT-3B-1*, and the first exon of *MFT-3B-2* was only present in K7.

### 2.4. Analysis of Differentially Expressed Proteins and GO Annotation for DEPs

In the differential expression proteins (DEPs) analysis, a total of 910 DEPs were obtained from the K0 vs. K7, W0 vs. W7, K0 vs. W0, and K7 vs. W7 groups (Appendix A). For the identified DEPs, the expression pattern was confirmed using volcano plot and heatmap analyses (Figure 5). In terms of up-/downregulated GO terms, there were 260 DEPs in the K0 vs. K7 group, 180 DEPs in the W0 vs. W7 groups, 179 DEPs in the K0 vs. W0 group, and 87 DEPs in the K7 vs. W7 groups, respectively. GO analysis in DEP had a relatively small number of annotations compared to the transcriptome. However, similar functional annotations were correlated with the results of the GO analysis, considering DEGs.

In the K0 vs. K7 group, the BP categories “response to abscisic acid”, “defense response”, “glycolytic process”, and “embryo development ending in seed dormancy” and response germination-related GO functions such as “response to water deprivation” and “lipid storage” were upregulated. In the CC category, “integral component of membrane”, “nucleus”, and “cytosol” accounted for most of the entries. In the MF category, “serine-type endopeptidase inhibitor activity” and “alditol: NADP+ 1-oxidoreductase activity” were upregulated. On the other hand, the downregulation of “ATP binding”, “3-hydroxypalmitoyl-[acyl-carrier-protein] dehydratase activity”, and “malate synthase activity” were annotated (Figure 6A). In the W0 vs. W7 group, the BP categories “response to abscisic acid”, “embryo development ending in seed dormancy”, “response to water deprivation”, “lipid storage”, and “lipid droplet formation” were upregulated, while “defense response”, “regulation of transcription, DNA-templated”, and “protein ubiquitination” were downregulated. In the CC category, only “membrane” was downregulated, while “nucleus”, “integral component of membrane”, and “cytosolic subunit” were upregulated. In the MF category, only the “structural constituent of ribosome” was upregulated, while GOs related to “ATP binding”, “protein kinase activity”, and “ADP activity” were downregulated (Figure 6B). In the K0 vs. W0 and K7 vs. W7 groups, small numbers of DEPs were annotated, compared to the K0 vs. K7 and W0 vs. W7 groups. Among the K0 vs. W0 group, in the BP category, “reproductive process” was confirmed to be upregulated, and “mitochondrial glycyl-tRNA aminoacylation” was downregulated. In the CC category, the annotated upregulated GO terms were “integral component of membrane” and “monolayer-surrounded lipid storage body”, while the annotated downregulated GO terms were “nucleus” and “plasma membrane”. In the MF category, upregulated functions such as “serine-type endopeptidase inhibitor activity” and “alpha-L-fucosidase activity” were annotated. On the other hand, downregulated functions were “ATP binding”, “DNA binding”, “chitinase activity”, and “protein kinase activity”. In particular, functions related to “ADP activity” were also annotated in the W0 vs. W7 group (Figure 6C). In the K7 vs. W7 group, “response to abscisic acid”, “cold acclimation”, and “response to water deprivation” functions were upregulated in the BP category. Only the “glyoxylate cycle” function, which is involved in the anabolic pathway of glucose production from fatty acids, was downregulated. The CC category was upregulated in “integral component of membrane”, while “nucleus” and “plasma membrane” were downregulated, in the same pattern as the K0 vs. W0 group. In the MF category, “ATP binding” and “malate synthase activity” were confirmed as downregulated functions (Figure 6D). The annotated results of the GO terms for DEPs are detailed in Appendix A.

### 2.5. Analysis of Correlation between RNA–Protein under PHS Treatment

The transcriptome and proteome crossover analysis under PHS conditions revealed that 35, 10, and 6 genes/proteins were cross-detected in the K0 vs. K7, W0 vs. W7, and K7 vs. W7 groups, respectively (Figure 7A–C, Table 1). However, there were no cross-detected DEGs/DEPs in the K0 vs. W0 group. The transcriptome–proteome comparison showed very low Pearson’s correlation coefficient values, with 924, 901, and 904 matched transcriptome/proteome genes in each group, respectively (Figure 7D–F). K0 vs. K7 showed a negative correlation (*r* = −0.37), while K0 vs. W0 showed a relatively weak correlation (*r* = −0.04). On the other hand, in the K0 vs. W7 group, a positive correlation was observed (*r* = 0.19). These results indicated that up-/downregulated DEGs were reversely translated at the protein level with down-/upregulated DEPs in K0 vs. K7, while the K7 vs. W7 group revealed a similar expression profile in transcription and translation (Figure 3A,B, Appendix A).

### 2.6. Validation of Selected DEGs via Quantitative Real-Time PCR

Validation of DEGs was performed by randomly selecting a total of 20 DEGs from PHS treatment between two cultivars (Appendix A). Six DEGs intersecting with DEPs were selected and verified using qRT-PCR (Appendix A). Additionally, nine DEGs related to the spliceosome and proteasome pathways were verified (Figure 8C,D). The relative expression levels via qRT-PCR were highly consistent with RNA-seq results. Moreover, the qRT-PCR validation demonstrated the reliability of the transcriptome and proteome profile results.

## 3. Discussion

Seed germination is affected by various morphological, ecological, and environmental factors. As seed germination is not processed uniformly, the expression during seed germination under water stress often shows remarkable variation. We compared spike morphology between the two cultivars. The comparison revealed that there was no significant difference in spike structure between Keumgang and Woori (Appendix A). In this study, the PHS-sensitive cultivar Keumgang and the PHS-tolerant cultivar Woori were used to investigate the PHS resistance mechanism. PHS resistance is endowed by a number of factors, including the morphological characteristics of the seed coat and spike structures, and by genetically controlled traits. Among those, seed dormancy is considered to be the major factor that determines PHS resistance [61]. In our previous study, we observed that the two cultivars showed remarkably different transcripts after ABA treatment, which is a seed dormancy hormone, and suggested there might be genetic differences in the control of seed dormancy [58]. Therefore, RNA-seq analysis was performed in order to identify PHS-related candidate genes and to investigate molecular responses under PHS. Additionally, differential expression analysis at the protein level and GO annotation were also performed. To improve the reliability of these results, we conducted a co-expression study between the transcriptome and proteome.

Song et al. [62] suggested that differences in water exposure are a major factor influencing transcript changes between treated samples. PHS stress is greatly affected by environmental circumstances; as such, it was difficult to maintain perfect reproducibility during the experiment. However, the modified sand bury method [63] that was utilized in this study showed significant PHS phenotype differences between the tolerant and sensitive cultivars (Figure 1). The conducted PHS experiment helped us to determine the phenotypic differences and differentially expressed genes between Keumgang and Woori under PHS. These results were similar to those presented in previous studies. Kim et al. [57] have conducted a PHS resistance profiling study on 28 Korean wheat varieties using artificial rainfall, the sand bury method, and a germination index (GI) test. Considering their results, we expected that our experiment would be highly reliable. Our modified sand bury method could be applied for PHS experiments, as well as artificial rainfall [58], the original sand bury method [63], and intended water loading [59], which are usually applied for PHS treatment.

GO and KEGG pathway annotation are useful methods for elucidating the functions of DEG groups in terms of the controlled vocabularies for gene functions or the interactions of DEGs in metabolic pathways, respectively [64]. Gene Ontology analysis can obtain functional annotations in the categories of gene-related biological processes, molecular functions, and the cellular components for individual genes, facilitating the study of gene function. The GO and KEGG annotation of the DEGs from K0, K7, W0, and W7 revealed that similar DEGs were represented during up- and downregulated conditions, in both the K0 vs. K7 and W0 vs. W7 groups (Figure 3A,B and Figure 4A,B). However, in terms of the downregulation of the two comparison groups, some functions confirming the difference in sensitivity and tolerance of the two cultivars under PHS were annotated. Representatively, ubiquitination-related GO, spliceosome, and abscisic acid were observed in K0 vs. K7. The Skp1-cullin 1-F-box (SCF) E3 ligase complex is the largest family of E3 ligases, among which the SCF E3 ubiquitin ligase promotes the degradation of cellular proteins such as signal transducers, cell cycle regulators, and transcription factors [65,66,67]. In particular, SCF-ubiquitin ligase is closely related to seed development and germination, and is also known as a regulator of the plant hormone auxin [68]. The plant genome bulk encodes the F-box protein (FBP), a substrate recognition sub-unit of the SKP1-CULLIN-F-box (SCF) ubiquitin ligase; TIR1 is one of the best-studied plant FBPs. TIR1 is associated with the CULLIN1 (CUL1) sub-unit and functions as a receptor for auxins [69]. In *Arabidopsis* over-expressing TIR1 gene, primary root elongation was suppressed and lateral root development was observed, while hypocotyl elongation was suppressed and dehydration was promoted, even in a dark environment, similar to the exogenous auxin response [68]. Interestingly, in the K7 vs. W7 group, the ubiquitin-related functions were in agreement with the results of Gray et al. [68], but showed an opposite profile to K0 vs. K7 (Figure 3A,D). Conversely, in the W0 vs. W7 group, germination-related functions such as oxidoreductase, UDP-glycosyltransferase, lipid metabolism, and glutathione tended to be suppressed. Glutathione and hydrogen peroxide activity have been shown to affect the dormancy of barley seeds and promote germination [70]. Reactive oxygen species (ROS) play an essential role in seed dormancy and germination. The maintenance of ROS homeostasis is a major function affecting seed germination. Glutathione peroxide mitigates oxidative damage by activating an internal antioxidant defense, which is partly responsible for the ROS erasing system [71]. Woori is a cultivar that is tolerant to germination; therefore, it can be interpreted that glutathione, which plays a main role in ROS detoxification, was suppressed at a higher level. The lipid metabolism process is a mechanism that accompanies germination and the degradation of storage lipid accumulation during seed germination, which is involved in the process of producing sucrose by lipid degradation in germination [72]. In addition, functions such as “UDP-glycosyltransferase activity” and “oxidoreductase activity” were linked with a high rank. Based on these results, it can be inferred that the GO functions of Woori involve various expressions of genetic factors that may be involved in PHS tolerance (Figure 3B). Furthermore, in K0 vs. W0, only a small number of GOs and no KEGGs were annotated, which indicates that significant expression differences were not observed between the two cultivars before PHS treatment (Figure 3C). However, a number of GOs and KEGGs were observed in K7 vs. W7, indicating that each cultivar had distinct expression levels in the same gene groups (Figure 3D and Figure 4C). In particular, “metal ion binding” was the most abundantly annotated in all groups, but it was only upregulated in the K7 vs. W7 group. The “metal ion binding” function has been scarcely researched in the context of wheat germination or PHS stress. It has only been studied regarding seed germination in *Lepidium sativum* L. belonging to the *Brassica* family, where it was shown that it had an inhibitory effect on the seed germination process, according to metal ion concentration [73]. Consideration of the GO functions with DEPs increased the reliability of the transcriptomic analysis. As the K0 vs. K7 group presented a negative R value (−0.37) in the Pearson correlation analysis, several GO functions were reversely classified between DEGs and DEPs, including ATP binding, DNA binding, protein kinase, chloroplast, cytosolic large complex, cytosol, and nucleus (Figure 3A and Figure 6A). However, K7 vs. W7 was not found to have correlated GO functions, even though it presented a positive R value (0.19); this was due to only approximately 2.6% of DEPs being classified with GO functions in the K7 vs. W7 group (Appendix A). Considering the small number of DEPs, the Pearson correlation might have been calculated with a low value. During the seed germination and breaking dormancy stage, transcription begins and the accumulation of transcriptomes first occurs. Only after sufficient transcription may proteins be detected at the seed germination stage. This may support why the DEGs were detected at a much higher rate than the DEPs, as shown in the Venn diagram (Figure 7). Few studies have co-profiled RNA and protein levels to assess the reliability of selected DEGs. Zhou et al. [74] carried out a 2-DE analysis in wheat callus after RNA-seq analysis in order to elevate the reliability for DEGs. Feng and Ma [56] have reported transcriptome and proteome profiling in bread wheat, where the transcriptome and proteome presented R correlation coefficients ranging from −0.007 to 0.081. Co-profiling between DEGs and DEPs shows the potential for increasing the reliability of the selected DEGs. Additionally, the DEP data were expected to play a complementary role in the functional annotation of selected DEGs under PHS treatment. Validation with qRT-PCR for “EM-like protein GEA1 (TraesCS1B02G237400)”, “Peroxides (TraesCS3A02G510600)”, and “Proteasome subunit alpha type (TraesCS7B02G170000)”, among the intersecting DEGs and DEPs in K7 vs. W7 group showed similar expression trends (Figure 7C and Appendix A). This approach could be helpful for understanding the mechanisms of PHS, germination, and seed dormancy.

Interestingly, “Spliceosome” and “Proteasome” were particularly highly expressed in K7 vs. W7 (Figure 8A,B). Alternative splicing is known as a mechanism regulating transcriptome and/or proteomic diversity, and is carried out through intron retention, exon skipping, and alternative 5’ or 3’ splice sites [75]. Generally, when one of the splicing sites is used, a single mRNA is generated from a multi-exon gene via constitutive splicing. However, it has been shown that alternative splicing is also frequently performed in multiple ways to produce multiple mRNAs of different sizes [75]. Thus, the transcript diversity is greatly increased, such that alternatively spliced transcripts contribute to the diversity and complexity of the proteome by encoding distinct proteins [76]. Proteins derived from alternative splicing may exhibit unintended expression, including additional functions, changed expression, and loss of function [76,77]. The role of these spliceosomes is involved in the response to various abiotic stresses and germination in plants [78]. Previous studies have shown that PHS tolerance is regulated by the alternative splicing of the viviparous gene (*Vp-1*), an important regulator of late embryonic development that is highly conserved in wheat and other species [78,79,80]. Zhang et al. [81] have revealed that alternative splicing hormone response genes are correlated with genes involved in protein biosynthesis and sugar metabolism genes in barley embryos, indicating that alternative splicing may play an important role in seed germination. In addition, the protein levels of ABI5, an ABA signaling component, have been associated with different PHS tolerance in sorghum [82]. In *Pyrus pyrifolia* (Burm.) Nak. and *Arabidopsis*, plant germination and flowering delay were observed to be affected by the structural change and functional inactivation of germination-related proteins due to substitutional conjugated isoforms [83]. In particular, pre-mRNA splicing in *Arabidopsis* is known to regulate ABA signaling, while SmEb—a key protein of the spliceosome component Sm—has been demonstrated to act as an ABA-positive regulator [84]. Figure 8C indicates upregulated expression for spliceosome-related transcriptome, including “sm-like protein LSM5 (TraesCS1B02G254400)”, “small nuclear ribonucleoprotein SmD1a-like (TraesCS2A02G331400)”, “splicing factor U2af large subunit B isoform X1 (TraesCS4D02G143400)”, and “pre-mRNA-processing protein 40C-like isoform X1 (TraesCS1B02G176100)” in Woori. Overall, these results suggested that the regulation of germination mechanism by the spliceosome might be more highly upregulated in the PHS-treated Woori than in Keumgang (Figure 8C). In particular, we observed different alternative splicing patterns in *MFT-3B-1* and *MFT-3B-2* between Keumgang and Woori (Appendix A). MOTHER OF FT AND TFL1 (*MFT*) is a member of the phosphatidylethanolamine-binding protein (PEBP) gene family in plants. *MFT* mainly identified effects on seed development and germination [85,86]. Quantitative trait loci (QTLs) related to PHS and seed dormancy have been reported at multiple locations on all chromosomes, and in particular, dominant QTLs have been identified on chromosomes 3AS and 4AL [87,88]. Among them, a candidate gene of QPhs.ocs-3A was identified as *MFT*, and germination delay due to SNP differences in the promoter region between varieties was confirmed [89]. In addition to *MFT-3A1*, *MFT-3B-1* was also identified as a locus related to PHS resistance through QTL analysis in a recent study [90]. The qRT-PCR results for alternative splicing also supported our hypothesis, as the selected DEGs were dominantly expressed in W7. An investigation into how the spliceosome regulates the *MFT* gene, and the role of differentially spliced *MFT* gene products might be key to understanding PHS resistance in wheat.

Plant responses in different developmental stages and abiotic stress are highly dependent on protein plasticity and regulation through various protein degradation pathways [91,92,93]. In plants, the 20S proteasome is responsible for the degradation of carbonylated proteins, whereas the 26S protease complex, belonging to the ubiquitin–proteasome pathway (UPP), is a representative component of the GA, ABA, and light signal transduction pathway [94]. The number of genes mediating UPP is tremendous in plant genomes, and different types of genes are involved in specific hormone signaling [95]. Furthermore, they can play opposite roles in the signal transduction of the same hormone (e.g., ABA), indicating the existence of complex relationships among hormones and various proteins [96]. In our study, proteasome-related pathways were significantly downregulated in Keumgang after PHS treatment, while no changes were observed in Woori (Figure 3 and Figure 4). It was also observed that all genes involved in the 26S proteasome were upregulated in W7 when compared to K7, such as the expression of “Proteasome sub-unit alpha type (TraesCS7B02G170000)”. These results might indicate the gap of expression of UPP under PHS treatment, leading to the difference in PHS sensitivity/tolerance. The roles of the UPP system in PHS have not yet been extensively investigated, and to the best of our knowledge, only a single related study has been reported in wheat. TaAIP3 and TaABI2 (ABI3-interacting proteins), which are part of the ubiquitin–proteasome system (UPS), induce poly-ubiquitination and proteasome degradation of target proteins, and their interaction with *TaVp1* led to differential expression levels in PHS-sensitive and -tolerant wheat cultivars [97]. Our research might provide fundamental knowledge toward understanding the role of UPS in PHS and PHS tolerance in wheat. The various questions related to the role of post-transcriptional mechanisms will continue to attract attention in the search for theories of the regulation of the complex processes in germination and dormancy.

## 4. Materials and Methods

### 4.1. Plant Materials and Treatments

This experiment was conducted using the Korean representative and PHS-sensitive cultivar Keumgang (K), and the PHS-tolerant cultivar Woori (W), both provided by the National Institute of Crop Science, RDA, Republic of Korea. Each cultivar was grown for 40 days after fertilization (DAF+40), based on the main tiller. PHS induction experiments were performed using a previously modified study method. The experimental period was 7 days in a growth control room. PHS induction conditions were created by filling a 765 × 485 × 105 mm plastic box with 4 L of golden vermiculite, and rainfall conditions were maintained for 12 h using a sprinkler that sprayed 2 L of mist per hour. Spikes of Keumgang (K0) and Woori (W0) before PHS induction were considered as control groups, while Keumgang (K7) and Woori (W7) on the 7th day of PHS induction were used as the experimental groups. Five independent whole spikes per cultivar were harvested before and after PHS induction treatment, which were stored at −70 °C until RNA extraction.

### 4.2. RNA Isolation, RNA-Seq Library Preparation, and Sequencing

Total RNA was extracted using a GeneAll Ribospin^TM^ Seed/Fruit Kit (GeneAll^®^ Biotechnology, Seoul, Korea), according to the manufacturer’s protocol. RNA purity and concentration were analyzed using a NanoDrop8000 spectrophotometer (Thermo Fisher Scientific, MA, USA). The total RNA integrity was measured using a Technologies 2100 Bio Analyzer (Agilent Technologies, Santa Clara, CA, USA), and the RNA quality criteria for library construction was adjusted (cut-off) to maintain an RNA integrity number (RIN) value of 7 or higher and an rRNA ratio of 1.5 or higher. A library for RNA-seq analysis was constructed using a TruSeq™ RNA library prep kit (Illumina, San Diego, CA, USA), and 100 bp paired-end sequencing was performed on the Illumina NovaSeq6000 platform (Illumina, San Diego, CA, USA).

### 4.3. Identification of DEGs Functional Annotation Analysis

Following the quality control of the sequencing data, the raw reads were mapped to the reference sequence of wheat RefSeq v1.0 in EnsemblPlant (https://plants.ensembl.org/Triticum_aestivum/Info/Index, accessed on 28 January 2022) using Tophat v2.0.13 (http://daehwankimlab.github.io/hisat2, accessed on 11 February 2022). For library normalization and dispersion estimation, geometric and pooled methods were applied, followed by DEG analysis using Cuffdiff (http://cole-trapnell-lab.github.io/cufflinks/cuffdiff/, accessed on 16 February 2022). Scatter and volcano plots were created for all DEGs. Scatter plots showed trends in the overall changes, and gene expression between comparative samples. Selected DEGs were limited to have high significance for candidate gene and functional annotation analysis. DEGs with statistically significant differences were compressed using the cut-off under the condition of >2-fold change and a *p*-value <0.005, and we considered the log_2_-fold change values of 1 and −1 as cut-off values for up-/downregulated genes, respectively. Amap, gplot, and heatmap analyses of R were performed, in order to identify the gene expression patterns of significantly expressed genes. Clustering analysis was performed using a compressed DEGs library, and expression similarity was analyzed using Pearson’s correlation coefficients. Further, the DEGs were functionally enriched in Gene Ontology (GO) and KEGG (Kyoto Encyclopedia of Genes and Genomes) terms. The Gene Ontology database (http://www.geneontology.org/, accessed on 1 April 2022) was used for the functional classification of all DEGs, according to the GO terminology [98], which was performed on DEGs with a *p*-value < 0.005. For KEGG pathway analysis, the IDs of DEGs were converted into the RefSeq assembly (GCF_018294505.1) gene IDs in the KEGG genome database (https://www.genome.jp/kegg-bin/show_organism?org=taes, accessed on 11 July 2022), and the annotated pathways were enriched and visualized using the cluster-Profiler package [99] in R v4.2 with a *p*-value < 0.05.

### 4.4. Protein Purification and Tandem Mass Tag (TMT) Labeling

Protein purification was performed using acetone precipitation from the wheat spikes used in the RNA-seq analysis. A sample of wheat spikes stored in SDS (10%) was added with 4 times the sample volume of ice-chilled acetone and stored at −20 °C. After sonication of the acetone-mixed sample, the tube was vortexed. The sample was incubated for 60 min at −20 °C and centrifuged for 10 min at 13,000–15,000× *g*. The supernatant was disposed carefully, in order to not dislodge the protein pellet. The acetone in the pellet was allowed to evaporate at room temperature for 30 min. Trypsin digestion was performed on the purified protein. An S-Trap mini spin column (Protifi, New York, NY, USA) was used to digest each sample, according to the manufacturer’s instructions. Three hundred milligrams of protein were reduced with 5 mM TCEP at 55 °C for 15 min and then alkylated with 20 mM of iodoacetamide at room temperature for 10 min in the dark. Phosphoric acid was added to the alkylated proteins at a final concentration of 1.2%, and six volumes of binding buffer (90% methanol; 100 mM TEAB; pH 7.1) were added to the acidified proteins. The sample was applied to the S-Trap column and centrifuged at 4000× *g* for 30 s, in order to trap protein. Afterwards, 400 µL of wash buffer (90% methanol and 100 mM TEAB; pH 7.1) were added three times to clean the protein. Finally, the protein was digested with trypsin gold at 37 °C overnight, at a protein-to-enzyme ratio of 10:1 (*w*/*w*). Digested peptides were eluted in three steps, using 80 µL of 50 mM TEAB in water, 0.2% formic acid in water, and 50% acetonitrile in water at 4000× *g* for 1 min. The pooled peptide solution was dried in a speed vacuum. Peptide samples were dissolved in 100 µL of 100 mM TEAB. A Ten-plex TMT kit was used to label the six samples. A total of 100 ug protein was labeled, according to the manufacturer’s protocol. TMT-labeled peptides were combined prior to offline basic reverse-phase liquid chromatographic (bRPLC) fractionation. The linear gradient was performed using Buffer A (10mM TEAB in water) and Buffer B (10 mM TEAB in 90% acetonitrile), and a total of 10 fractions were analyzed using an LC-MS/MS system.

### 4.5. LC-MS/MS Analysis

LC-MS/MS analysis was performed on 10 samples dissolved in 0.1% formic acid using an UltiMate 3000 RSLCnano system and an Orbitrap Eclipse Tribrid mass spectrometer (Thermo Fisher Scientific, Waltham, MA, USA). Using an auto sampler, the sample solution was loaded onto a C18 trap column (Acclaim PepMap™ 100, 75 μm × 2 cm, Thermo Fisher Scientific, Waltham, MA, USA) and concentrated on the trap column for 9.5 min at a flow rate of 4 μL/min. The mobile phase consisted of 99.9% water (A) and 99.9% ACN (B), each containing 0.1% formic acid. The LC gradient was run starting with 5% B for 10 min, 13% B for 40 min, 25% B for 65 min, and 95% B for 5 min. Thereafter, it was held at 95% B for 5 min and at 5% B for an additional 1 min. The column was re-equilibrated to 5% B for 14 min before the next run, and a voltage of 1900 V was applied to generate ions. During chromatographic separation, an Orbitrap Eclipse Tribrid mass spectrometer (Thermo Fisher Scientific, Waltham, MA, USA) was operated in data-dependent mode with automatically switching between MS1 and MS2. Full-scan MS1 spectra (400–2000 m/z) were acquired by Orbitrap, with a maximum ion implantation time of 100 ms at a resolution of 120,000, and an automatic gain control (AGC) target value of 4.0 × 10^5^. MS2 spectra were acquired using an Orbitrap mass spectrometer at a resolution of 30,000 using HCD (36% normalized collision energy, maximum ion implantation time of 50 ms, AGC target value of 5.0 × 10^4^). Previously fragmented ions were excluded for 30 s within 10 ppm.

### 4.6. Proteome Search and Bioinformatics Analysis

All MS raw files were converted into mzML and ms2 file formats using the MSConvert software (version 3.0.20033, Stanford University, Stanford, CA, USA; https://proteowizard.sourceforge.io/tools/msconvert.html, accessed on 10 June 2022). In order to determine the wheat proteome, we downloaded a protein FASTA file of *Triticum aestivum* from Uniprot (http://uniprot.org, accessed on 28 June 2022), including 169,427 reviewed (Swiss-Prot) and unreviewed (TrEMBL) proteins entries, and we generate a proteome search database with reversed sequences and contaminants in the Integrated Proteomics Pipeline (IP2, version 5.1.2, Integrated Proteomics Applications Inc., San Diego, CA, USA). The proteome search from 20 ms2 files was performed with IP2 and its following parameters: Both ms1 and ms2 search tolerances were allowed within 20 ppm, a peptide length of 6 or more amino acids, static modifications of 229.1629 at the N-terminal and Lysine (K) and 57.02146 at Cysteine (S), a variable modification of 15.9949 at Methionine (M), Trypsin digestion enzyme, and a maximum allowable mis-cleavage of 2. The proteome search results were evaluated, considering a false discovery rate (FDR) at spectra and a protein level of less than 1.0%, using IP2 and Proteininferencer (version 1.0, Integrated Proteomics Applications Inc., San Diego, CA, USA), respectively. Protein quantification and statistical analysis for the discovery of DEPs was performed from the ms2 files with TMT reporter ions using an in-house program coded using Python 3.8, where a *t*-test and Pearson’s correlation analysis between the comparison samples was performed using the scikit-learn (version 0.23.2, accessed on 26 July), Scipy (version 1.6.0, accessed on 26 July 2022), and statsmodels (version 0.12.1, accessed on 26 July 2022) Python libraries. DEPs with statistically significant differences were compressed via cut-off under the condition of a 1.5-fold change using the log_2_ change value and a *p*-value of <0.05. Gene Ontology (GO) analysis with DEPs in each comparison was performed with the GO information from the Uniprot database, where Fisher’s exact test of a two-sided hypothesis was conducted. Finally, the sum of difference values for each GO term with its corresponding proteins was calculated using an in-house program coded in Python 3.8.

### 4.7. Validation Gene Expression Analysis

Quantitative real-time PCR (qRT-PCR) was performed using a QIAGEN Rotor-Gen Q (QIAGEN, Hilden, Germany) with a Rotor-Gen SYBR Green PCR kit (QIAGEN, Hilden, Germany) according to the qRT-PCR process described by Kim et al. [100]. The selected DEG- and DEP-specific primer pairs were designed using Primer3 (https://www.primer3plus.com/) software, and the qRT-PCR conditions followed the thermal cycling conditions suggested by the manufacturer. All experiments were performed in three biological and technical replicates, and relative transcription levels were normalized using *TaActin*. Relative expression levels were analyzed via normalization using the 2^−ΔΔCt^ method [101]. Further information for the validated primer sequences for DEGs and DEPs is provided in more detail in Appendix A.

## 5. Conclusions

The PHS trait in wheat has been studied for a long time; however, the mechanism underlying PHS remains unclear. We conducted a PHS treatment experiment on two wheat cultivars, Keumgang and Woori, and then analyzed how PHS sensitivity/tolerance relates to transcriptomic expression. The expression of DEGs/DEPs between the two cultivars under PHS treatment was simultaneously profiled. In addition, the pathways for the related functions were identified and screened through KEGG pathway analysis. DEG/DEP functional annotations for each comparison group showed similar expressions, where functions related to ”spliceosome” and ”proteasome” were increased by more than 70% in Woori (W7) compared to Keumgang (K7). The results reported in this study suggest the possibility of the difference in PHS sensitivity/tolerance being related to the ”spliceosome” and “ubiquitin-proteasome”, which have various effects in response to abiotic stresses. Moreover, co-profiling analysis between the transcriptome and proteasome enhanced the reliability of our transcriptome study. These results can be used as fundamental information to further improve our understanding of seed germination and dormancy mechanisms in wheat.

## Figures and Tables

**Figure 1 plants-11-02807-f001:**
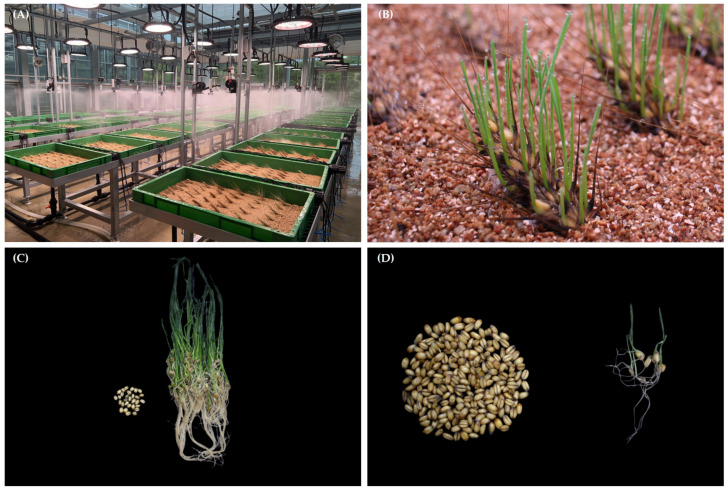
Phenotypic changes were induced in Keumgang and Woori spikes under pre-harvest sprouting (PHS) treatment. (**A**) The experiment was carried out for 7 days in an artificial growth control room to maintain optimal PHS induction conditions. (**B**) Wheat spikes on the 5th day of PHS induction; active germination is progressing by treatment. (**C**) Keumgang phenotype assay 7 days after PHS induction. (**D**) Woori phenotype assay 7 days after PHS induction.

**Figure 2 plants-11-02807-f002:**
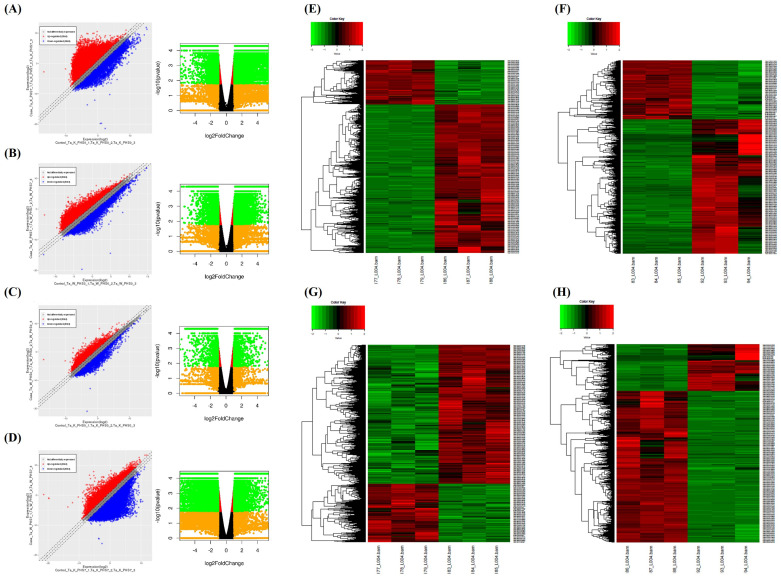
Scatter plot, volcano plot, and heatmap analysis of differentially expressed genes (DEGs) in K0 vs. K7, W0 vs. W7, K0 vs. W0, and K7 vs. W7 groups. (**A**) Scatter plot and volcano plot of the K0 vs. K7 group, (**B**) scatter plot and volcano plot of the W0 vs. W7 group, (**C**) scatter plot and volcano plot of the K0 vs. W0 group, (**D**) scatter plot and volcano plot of the K7 vs. W7 group, (**E**) heatmap analysis for DEGs in the K0 vs. K7 group, (**F**) heatmap analysis for DEGs in the K0 vs. K7 group, (**G**) heatmap analysis for DEGs in the K0 vs. W0 group, and (**H**) heatmap analysis for DEGs in the K7 vs. W7 group. The *X*-axis and *Y*-axis of the scatter plot represent a control group and a case group, respectively, and the value of the axis is the average of the values normalized on a log_2_ scale. Red dots indicate more than a doubling of upregulated DEGs, and blue dots indicate more than a doubling of downregulated DEGs. Gray dots indicate no differential expression. The volcano plot is expressed as a *p*-value derived from the log_2_ fold change of the expression value for each comparison group and the average comparison between the two groups. The *X*-axis represents log_2_ fold change; *Y*-axis represents −log_10_ *p*-value. Heatmap analysis was performed on differentially expressed genes in each comparison group, and the expression was adjusted to *p*-value < 0.005. Red indicates upregulation and green indicates downregulation.

**Figure 3 plants-11-02807-f003:**
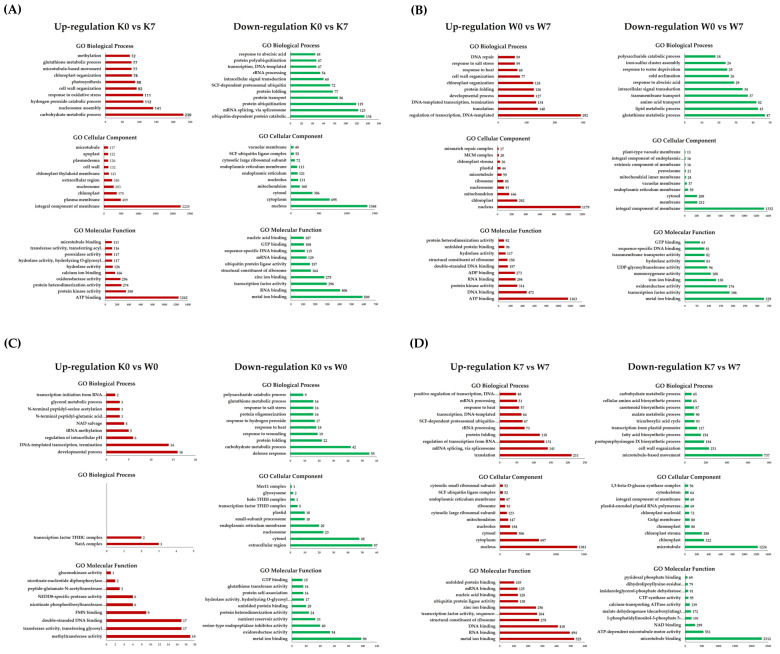
Gene Ontology analysis in DEGs for the K0 vs. K7, W0 vs. W7, K0 vs. W0, and K7 vs. W7 groups. Upregulation (red) and downregulation (green) are shown for each group, and GO terms belonging to the categories of biological processes, cellular components, and molecular functions are indicated. The *X*-axis represents the number of annotated DEGs, and the *Y*-axis lists the 10 most annotated GO functions by category. (**A**) Annotated up-/downregulated GO terms in K0 vs. K7. (**B**) Annotated up-/downregulated GO terms in W0 vs. W7. (**C**) Annotated up-/downregulated GO terms in K0 vs. W0. (**D**) Annotated up-/downregulated GO terms in K7 vs. W7. The volcano plot is expressed as a *p*-value derived from the log_2_ fold change of the expression value for each comparison group and the average comparison between the two groups. The *X*-axis represents log_2_ fold change; the *Y*-axis represents −log_10_ *p*-value. Heatmap analysis was performed on differentially expressed proteins in each comparison group.

**Figure 4 plants-11-02807-f004:**
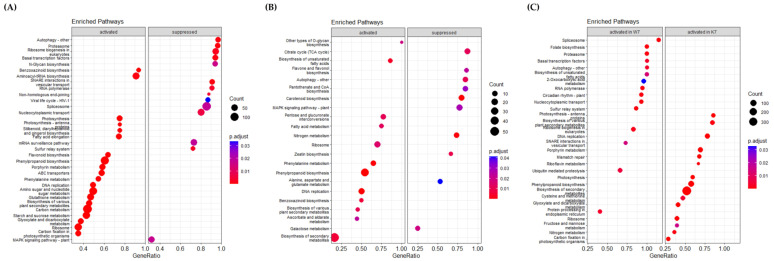
KEGG (Kyoto Encyclopedia of Genes and Genomes) pathway analysis for PHS-treated K0 vs. K7, W0 vs. W7, and K7 vs. W7 groups. GeneRatio on the *X*-axis is the ratio of the enhancement targets in the pathway to all targets annotated with KEGG, and the *Y*-axis lists the relevant functions. Each control group was divided into “activated” and “suppressed”. The size of the dot indicates the number of targets that could be annotated in the KEGG database, and the significance of the analysis was indicated by the p.adjust value. (**A**) KEGG analysis associated with identified DEGs in the K0 vs. K7 group. (**B**) KEGG analysis associated with identified DEGs in W0 vs. W7. (**C**) KEGG analysis associated with identified DEGs in the K7 vs. W7 group.

**Figure 5 plants-11-02807-f005:**
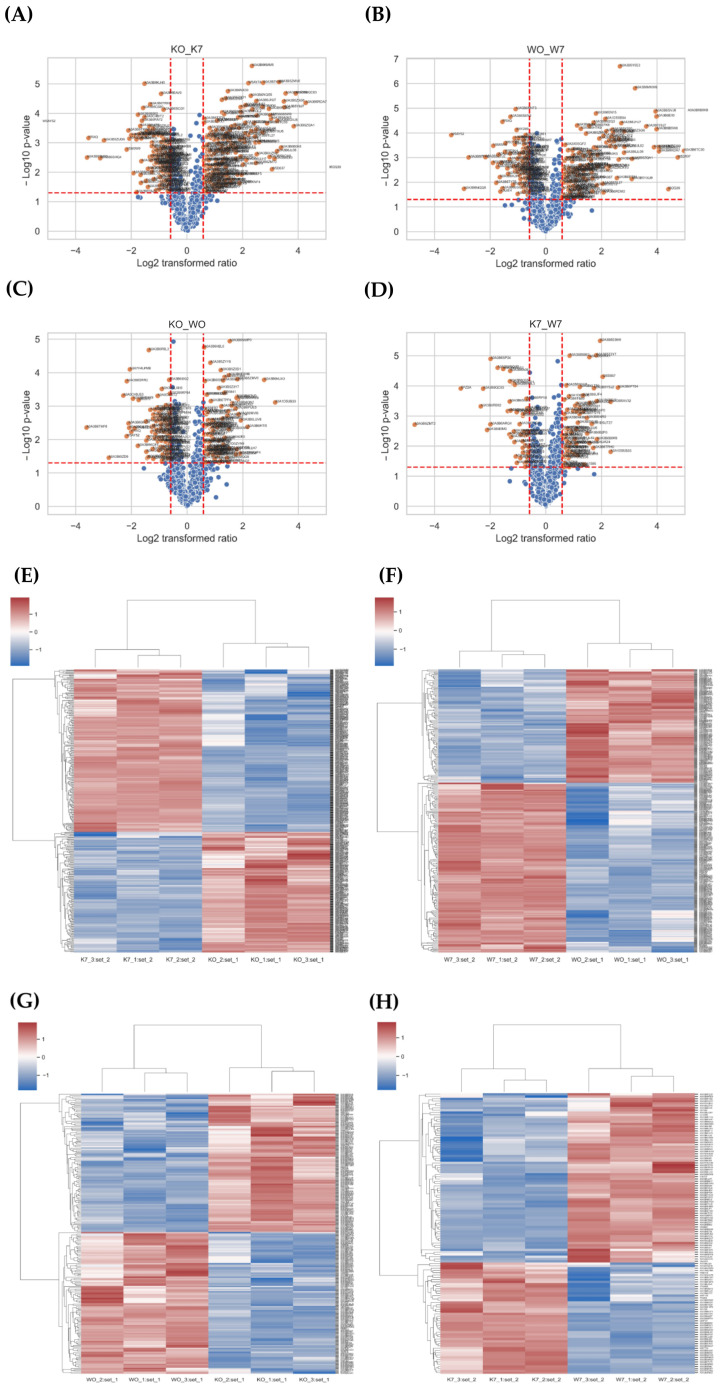
Volcano plot and heatmap analysis of differentially expressed proteins (DEPs) in K0 vs. K7, W0 vs. W7, K0 vs. W0, and K7 vs. W7 groups. (**A**) Volcano plot of the K0 vs. K7 group, (**B**) volcano plot of the W0 vs. W7 group, (**C**) volcano plot of the K0 vs. W0 group, (**D**) volcano plot of the K7 vs. W7 group, (**E**) heatmap analysis for DEGs in K0 vs. K7 group, (**F**) heatmap analysis for DEGs in the W0 vs. W7 group, (**G**) heatmap analysis for DEGs in the K0 vs. W0 group, and (**H**) heatmap analysis for DEGs in the K7 vs. W7 group.

**Figure 6 plants-11-02807-f006:**
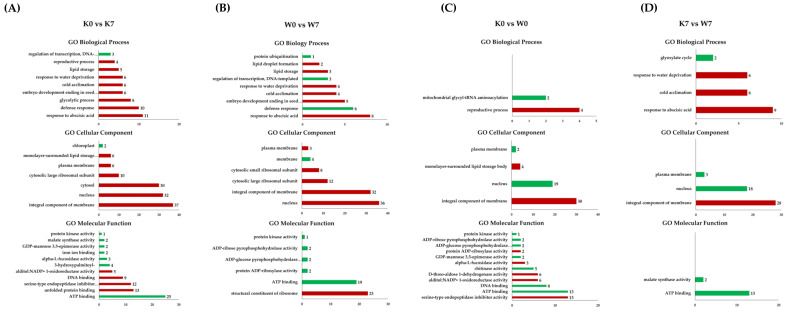
Gene Ontology comparisons with respect to cultivars and duration of PHS treatment in proteome analysis. GO analysis was performed on DEPs identified in the K0 vs. K7, W0 vs. W7, K0 vs. W0, and K7 vs. W7 groups, and GO terms belonging to the categories of biological processes, cellular components, and molecular functions are indicated. The upregulated (red) and downregulated (green) terms were divided by group. The *X*-axis represents the number of annotated DEPs, and the *Y*-axis lists GO functions by category. The number of GOs performed with DEPs was comparatively smaller than that of the GOs performed with DEGs, and so both the up- and downregulated entries are displayed in one graph. (**A**) Annotated GO terms in K0 vs. K7, (**B**) annotated GO terms in W0 vs. W7, (**C**) annotated GO terms in K0 vs. W0, and (**D**) annotated GO terms in K7 vs. W7.

**Figure 7 plants-11-02807-f007:**
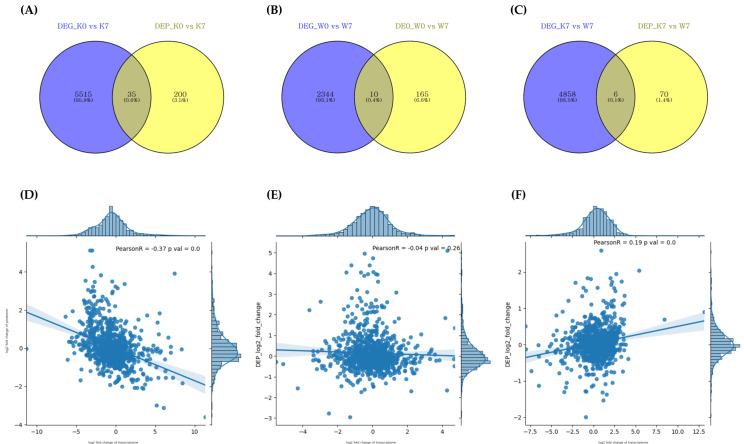
Correlation analysis between proteome and transcriptome by DEG/DEP expression correlation Venn diagram (**A**–**C**). DEG_comparison group and DEP_comparison group indicating DEGs/DEPs identified at *p*-value < 0.005, respectively. (**A**) Intersecting DEGs/DEPs for K0 vs. K7, (**B**) intersecting DEGs/DEPs for W0 vs. W7, and (**C**) intersecting DEGs/DEPs for K7 vs. W7. Correlation analysis between identified DEGs/DEPs in each comparison group (**D**–**F**). The *X*-axis is the number of undistributed proteins, and the *Y*-axis is the number of undistributed genes. The correlation coefficient and *p*-value between the transcriptome and the proteome are also shown, and each dot represents a DEG/DEP. (**D**) Correlation of DEGs/DEPs in K0 vs. K7, (**E**) correlation of DEGs/DEPs in W0 vs. W7, and (**F**) correlation of DEGs/DEPs in K7 vs. W7.

**Figure 8 plants-11-02807-f008:**
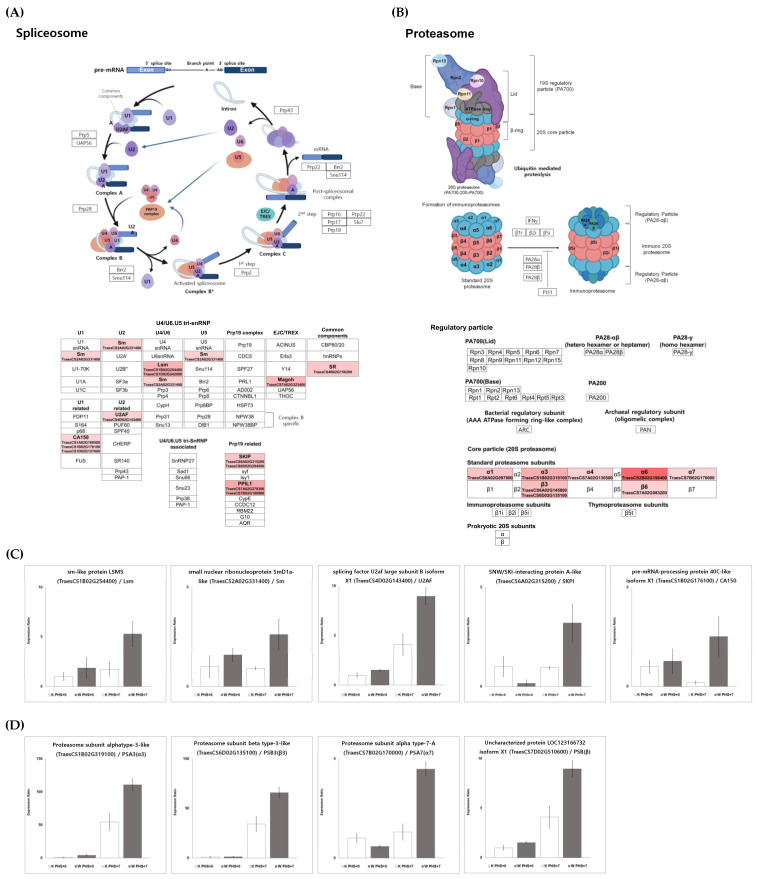
The relative expression profiles and validation of spliceosome and proteasome. The spliceosome pathway was reconstructed based on the KEGG database using Bioreder software. The expression of transcriptome related to the upregulated pathway is represented by red intensity. (**A**) The spliceosome pathway and its related genes represented using KEGG pathway analysis, (**B**) the proteasome pathway and its related genes represented using KEGG pathway analysis, (**C**) validation of genes related to the spliceosome pathway by qRT-PCR, and (**D**) validation of genes related to the proteasome pathway by qRT-PCR.

**Table 1 plants-11-02807-t001:** A list of genes that can be simultaneously identified in DEGs and DEPs selected for each group.

Gene ID	Protein ID	Description	Gene* log2 (fold_change)	Protein* log2 (fold_change)	Gene Functional Classification
**DEGs + DEPs K0 vs. K7**					
TraesCS6D02G040600	A0A3B6QC63	Histone H2B	7.454697426	3.913866667	protein heterodimerization activity
TraesCS1D02G087600	A0A3B5ZPT5	Carboxypeptidase	4.256589483	−0.844066667	serine-type carboxypeptidase activity
TraesCS5B02G078300	A0A3B6LFH5	Pyrophosphate--fructose 6-phosphate 1-phosphotransferase subunit alpha	3.945638767	1.7209	ATP binding, phosphofructokinase activity
TraesCS5A02G238100	A0A3B6KJH0	Alpha-amylase	3.945638767	−1.524566667	carbohydrate metabolic process, alpha-amylase activity
TraesCS4D02G022300	A0A3B6JCR8	Dirigent protein	3.697823539	−1.883066667	apoplast
TraesCS7D02G369400	A0A3B6TMP4	Peroxidase	3.494350899	−2.1036	hydrogen peroxide catabolic process, response to oxidative stress
TraesCS4A02G246100	A0A0C4BJE5	Serine hydroxymethyltransferase	3.352185251	−2.068633333	carbon metabolic process, pyridoxal phosphate binding
TraesCS4B02G069300	A0A0C4BJE5	Serine hydroxymethyltransferase	2.681982576	−2.068633333	carbon metabolic process, pyridoxal phosphate binding
TraesCS4D02G068100	A0A0C4BJE5	Serine hydroxymethyltransferase	2.371852235	−2.068633333	carbon metabolic process, pyridoxal phosphate binding
TraesCS7A02G172700	A0A3B6RBU0	NAD(P)H dehydrogenase C1	2.286001127	−0.597666667	chloroplast
TraesCS7D02G030700	Q8RW00	Glutathione transferase	1.702418296	−1.2913	cytoplasm
TraesCS2D02G075000	A0A1D5UXN7	Amine oxidase	1.338600011	1.4274	amine oxidase activity, amine metabolic process
TraesCS7B02G315600	A0A3B6SQU7	Dolichyl-diphosphooligosaccharide--protein glycosyltransferase 48 kDa subunit	1.210570417	0.941966667	oligosaccharyltransferase complex
TraesCS2A02G277100	A0A3B6AYU7	Peptidylprolyl isomerase	1.158355889	−1.445033333	cytoplasm
TraesCS4A02G445300	D3K1B4	Ozone-responsive stress-related protein	1.111810845	−1.262133333	response to ozone
TraesCS3A02G521500	P29557	Eukaryotic translation initiation factor 4E-1	−1.085773524	0.854366667	RNA 7-methylguanosine cap binding
TraesCS2B02G567600	A0A3B6CFS9	Superoxide dismutase	−1.08821619	1.462	mitochondrion
TraesCS2B02G148400	A0A3B6C126	Cysteine proteinase inhibitor	−1.170117935	1.340766667	cysteine-type endopeptidase inhibitor activity
TraesCS1B02G142000	A0A3B5YUP3	Phosphotransferase	−1.543828061	0.699933333	cytosol, glucose binding
TraesCS1D02G369300	A0A024CKY0	LEA protein	−1.580477896	2.0741	protein targeting
TraesCS1B02G237400	A0A3B5YXU6	Em-like protein GEA1	−1.683504359	2.719933333	cytosol, response to abscisic acid
TraesCS7A02G558300	A0A3B6RT93	4-Hydroxy-4-methyl-2-oxoglutarate aldolase	−1.684719777	−1.087933333	Metal-binding, metal ion binding
TraesCS1D02G101700	A0A3B5ZQA1	TSPO	−1.884546698	3.839566667	integral component of membrane
TraesCS7A02G070900	A0A3B6R8P8	Peroxidase	−2.316372851	2.466533333	metal ion binding
TraesCS2D02G114100	A0A3B6DAQ7	Glycosyltransferase	−2.522142222	1.4226	UDP-glycosyltransferase activity
TraesCS5D02G379300	A0A0H4MAT1	Dehydrin	−2.5490366	1.051233333	cytosol, cold acclimation
TraesCS2B02G402500	W5B7W5	Caleosin	−2.72237768	2.053666667	calcium ion binding
TraesCS2B02G384600	W5B8D6	Caleosin	−2.789346082	2.7154	calcium ion binding
TraesCS2A02G385600	W5AY74	Caleosin	−2.841920013	2.209666667	calcium ion binding
TraesCS2D02G382300	A0A1B5GE57	Caleosin	−3.018950089	0.895933333	calcium ion binding
TraesCS5B02G046000	D9ZLW0	Outer membrane channel protein OEP16-2	−3.060325217	1.923266667	integral component of chloroplast outer membrane
TraesCS5D02G188400	A0A3B6MQN2	Oleosin	−3.167389471	1.485	lipid storage, monolayer-surrounded lipid storage body
TraesCS3D02G467300	A7UME2	Xylanase inhibitor 725ACCN	−3.297615658	1.548766667	xylan catabolic process
TraesCS5B02G181700	A0A3B6LLH7	Oleosin	−3.484907598	2.133566667	lipid storage, monolayer-surrounded lipid storage body
TraesCS6A02G077000	Q4W6G2	Xylanase inhibitor XIP-III	−3.778453786	0.982	xylan catabolic process
**DEGs + DEPs W0 vs. W7**					
TraesCS5B02G078300	A0A3B6LFH5	Pyrophosphate--fructose 6-phosphate 1-phosphotransferase subunit alpha	4.161210781	1.848733333	ATP binding, nucleus
TraesCS2A02G277100	A0A3B6AYU7	Peptidylprolyl isomerase	1.915260225	−1.1109	peptidyl-prolyl cis-trans isomerase activity
TraesCS5D02G182500	A0A3B6MRT6	Phosphoenolpyruvate/phosphate translocator 1, chloroplastic	1.226990527	0.993233333	chloroplast membrane
TraesCS2B02G350500	A0A3B6C8B8	ATP-dependent 6-phosphofructokinase	−1.12833302	−1.382833333	magnesium
TraesCS2B02G402500	W5B7W5	Caleosin	−1.171315189	1.729333333	calcium ion binding
TraesCS2B02G429100	A0A3B6CB61	Glutathione peroxidase	−1.271800178	−0.7268	cytosol
TraesCS1D02G327200	A0A3B5ZYU5	Thioredoxin	−1.486027075	−1.1557	thioredoxin peroxidase activity
TraesCS7A02G558300	A0A3B6RT93	4-Hydroxy-4-methyl-2-oxoglutarate aldolase	−1.680230898	−0.745633333	metal-binding
TraesCS3D02G467300	A7UME2	Xylanase inhibitor 725ACCN	−2.989213884	2.6337	aspartic-type endopeptidase activity
TraesCS1D02G369300	A0A024CKY0	LEA protein	−3.620246235	2.238733333	protein targeting
**DEGs + DEPs K7 vs. W7**					
TraesCS4B02G225400	Q9SBB7	Chloroplast small heat shock protein	5.40441398	2.0434	rRNA processing
TraesCS1B02G237400	A0A3B5YXU6	Em-like protein GEA1	3.420870011	1.094333333	response to abscisic acid, cytosol
TraesCS3A02G510600	A0A3B6ESH8	Peroxidase	2.705166293	1.952333333	metal ion binding
TraesCS4A02G332100	A0A3B6HYW3	Nascent polypeptide-associated complex subunit beta	1.164770655	0.752066667	cytosol
TraesCS7B02G170000	A0A1D6CXF2	Proteasome subunit alpha type	1.015394237	2.3666	nucleus, cytoplasm
TraesCS2A02G571300	A0A3B6B862	Peroxidase	1.015085068	0.8543	metal ion binding
TraesCS5A02G142200	A0A3B6KDG2	Homoserine dehydrogenase	−1.226585437	1.568266667	homoserine dehydrogenase activity, homoserine metabolic process

* Red represents upregulation and green represents downregulation.

## Data Availability

The RNA-seq data of the present investigation were submitted to the NCBI SRA database under the bioproject ID PRJNA862687.

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
