# Peer review of "Transcriptome and Proteome Co-Profiling Offers an Understanding of Pre-Harvest Sprouting (PHS) Molecular Mechanisms in Wheat (Triticum aestivum)"

_plants, 2022, doi:10.3390/plants11212807_

Round 1
Reviewer 1 Report
Pre-harvest sprouting is a global problem for cereals, including wheat, barley, sorghum, rice, and maize. Though many genes or QTL related to PHS resistance were detected in wheat, more stable candidate genes are needed. Combining transcriptome and proteome data analysis may show some critical genes/proteins associated with PHS. However, they may be the results instead of the reasons for PHS resistance. The authors concluded that alternative splicing and ubiquitin-proteasome support the regulation of germination and seed dormancy by the NGS and proteome data. But this MS should show more solid evidence for its publication.
Line 131: Why was the experiment under 28/15? The high temperature will reduce seed dormancy during grain development.
Line 148-150: How many reads were mapped? There were two conflicting numbers.
Line 171-172: The DEGs between two cultivars should differ because of their variant genetic background.
Line 181-308: This redundancy part was described in Table3 and Figure3. The readers can obtain these numbers from the table and figure.
Line 384: The combination of the results of DEGs and DEPs should be the most important data for this study. But the authors only listed the numbers and made a simple statistical analysis. The genes/proteins should be analyzed in detail and shown in the main text.
Line 389: Several PHS-related genes were reported in wheat. But they were not in the list of figures S2, S3, and table S6.
Author Response
Response to reviewers’ comment:
We would like to thank the guest editor and reviewers for the constructive comments that helped to significantly improve the enclosed revised version of our manuscript. We have reviewed and agreed with all the comments and suggestions made by the reviewers, and have addressed each comment separately. Additionally, English correction of the entire manuscript for grammar and sentence form has been completed by a native speaker. Please find below a detailed and itemized list of our responses to the points raised by the reviewers. Our responses to the reviewers’ questions/suggestions are highlighted in the revised manuscript in yellow.
Comments and Suggestions for Authors
Reviewer #1
Comments to the Author
Pre-harvest sprouting is a global problem for cereals, including wheat, barley, sorghum, rice, and maize. Though many genes or QTL related to PHS resistance were detected in wheat, more stable candidate genes are needed. Combining transcriptome and proteome data analysis may show some critical genes/proteins associated with PHS. However, they may be the results instead of the reasons for PHS resistance. The authors concluded that alternative splicing and ubiquitin-proteasome support the regulation of germination and seed dormancy by the NGS and proteome data. But this MS should show more solid evidence for its publication.
Answers: The authors appreciate and agree with the reviewer’s valuable comments. The authors have performed additional experiments and data analysis for solid evidence. In order to obtain evidence for the regulation of germination and seed dormancy by alternative splicing and ubiquitin–proteasome, the authors conducted Integrative Genomics Viewer (IGV) analysis with the raw data of the read map for the PHS related genes, Vp-1, DOG1, and MFT. Among the genes, MFT3B-1 and MFT3B-2 especially represented differential alternatively spliced transcripts (Supplementary Figure S4). The alternative splicing pattern of two genes was similar in both species before PHS treatment (K0 and W0) in MFT-3B-1 and MFT3B-2. However, after PHS treatment, the second and the third exon of MFT3B-1 was absent in K7 but present in W7, while the fourth exon was absent in W7 but present in K7. Moreover, the first exon in MFT3B-2, was lost in W7, which implies that there might be differences in how the spliceosome acts between the two species. Additionally, the authors performed qRT-PCR for selected DEGs, as presented in Figure 8C. The selected DEGs from alternative splicing and ubiquitin–proteasome were displayed higher expression in W7 than K7 (Figure 8C, D). Authors also performed volcano plot and heatmap analysis of DEPs for alternative splicing in K7 vs W7 group (Supplementary Figure S4). A total of 42 alternative splicing unique variants were identified in W7. The results of IGV, qRT-PCR, ands DEP variants lead us to be convinced of our hypothesis. These additional experiments and analysis could provide indirect evidences for the regulation of alternative splicing and ubiquitin–proteasome under PHS treatment. The authors have added these results into the Results section and Discussion section. [lines 258-269, 510-530, 557 and 714-715]
Minor comments:
- Line 131: Why was the experiment under 28/15? The high temperature will reduce seed dormancy during grain development.
Answers: In this study, we used two Korean cultivars, Keumgang and Woori, which were optimized for the Korean growth environment. Because wheat PHS in Korea usually occurs in June, we performed PHS experiments using actual wheat growth conditions. According to the Korea Meteorological Administration, the average temperature in June is 27.52/18.63 ℃. This is the reason for why we performed the PHS experiment at 28/15℃. The following table is a summary for the average high temperatures and low temperatures in Korea in June from 2020 to 2022.
|
Years |
Average high Tm. |
Average low Tm. |
|
June. 2020 |
27.74 |
18.53 |
|
June. 2021 |
27.18 |
18.14 |
|
June. 2022 |
27.64 |
19.24 |
|
Average |
27.52 |
18.63 |
Korea Meteorological Administration (KMA)
- Line 148-150: How many reads were mapped? There were two conflicting numbers.
Answers: The total number of mapped reads in the four samples was 69,035,752 reads. Among them, a total of 57,708,213 reads were mapped, showing 83.56% mapping coverage (Supplementary Table S1). [lines 150-152]
- Line 171-172: The DEGs between two cultivars should differ because of their variant genetic background.
Answers: The authors agree with the reviewer’s comment and the sentences have been removed. [line 160]
- Line 181-308: This redundancy part was described in Table3 and Figure3. The readers can obtain these numbers from the table and figure.
Answers: The authors agree with the reviewer's comments and Section 2.3 has been summarized. Gene Ontology results were mentioned as the main function of the manuscript, and an explanation was carried out over a larger range than before. [lines 181-226]
- Line 384: The combination of the results of DEGs and DEPs should be the most important data for this study. But the authors only listed the numbers and made a simple statistical analysis. The genes/proteins should be analyzed in detail and shown in the main text.
Answers: Supplementary Table S6 was rearranged as Table 1 in the main text. In the renamed Table 1, gene fold-change, protein fold-change, and gene functional classification were added. Color gradients to represent the expression values were added to the columns for gene fold-change and protein fold-change. The intensity of red or green stand for the two-fold change expression level. [line 356]
- Line 389: Several PHS-related genes were reported in wheat. But they were not in the list of figures S2, S3, and table S6.
Answers: The authors agree with the reviewer's comments. From the DEGs analysis, we identified DOG1and Vp-1 as known PHS resistance genes in K7 vs. W7. However, a representative PHS-related gene could not be identified from overlapping genes between DEGs and DEPs. We think that although DOG1 and Vp-1 were differentially expressed in K7 vs. W7, DOG1 and Vp-1 were not detected in differential protein expression in K7 vs. W7. The authors also performed qRT-PCR to investigate the expressions of DOG1 and Vp-1, and added the expression validation results of the PHS-related genes to Supplementary Figure 2.

Reviewer 2 Report
Summary
This is a valuable study for wheat pre-harvest sprouting (PHS) resistance. Wheat PHS resistance is very important for wheat yield and end-use quality, especially in rain-fed areas. The authors performed RNA-sequencing and proteomic analysis for two wheat cultivars with very different sprouting rate at day 0 and day 7. DEGs and DEPs were identified from the transcriptomic and proteomic data, and gene ontology enrichment analysis for DEGs and DEPs elucidated pathways that contribute to PHS resistance. This study shed lights on biological pathways that associated with PHS resistance and provided candidate genes for future studies to understand the molecular mechanism for PHS resistance.
Major issues
1. The research used PHS susceptible lines “Keumgang” and PHS resistance line “Woori” to study their transcriptomic and proteomic differences at day 0 and day 7 of PHS inducement. The K7 sample was representative of sprouting process while the W7 sample was not (very few seeds germinated). Therefore, DEGs and DEPs identified in the K0 vs K7 comparison but not in the W0 vs W7 comparison should be more likely to be associated with sprouting. On the contrary, DEGs and DEPs identified in the W0 vs W7 comparison but not in the K0 vs K7 comparison should be more likely to be associated with sprouting resistance.
2. The 2.3 section in results is very long and descriptive, and all information can be found in Figure 3, which is a bit redundant. As a reader, I would like to see a summary of the GO enrichment analysis, i.e. what are the similarities across the DEGs in the four comparisons, what are the unique pathways in some comparisons, etc.
3. The common genes between DEGs and DEPs seem to be very important. In the results, I suggest the authors at least provide a brief classification of those gene functions. Providing gene numbers is not very informative in the context.
4. PHS resistance is affected not only by seed dormancy, but also by seed coat and spike structure. Genetic differences could not be identified in this study because the transcriptomic and proteomic data in this study could not capture the seed and spike development processes, which could be some limitations that need be mentioned in discussion.
5. It is very interesting that alternative splicing seems to play a role in responding to the abiotic stresses and germination. I wonder if it is possible to check if alternative splicing was observed in some of target genes (e.g. Vp-1) of the spliceosome proteins using the current transcriptomic data and proteomic data?
Minor issues
1. Please provide full names for abbreviations as they are used first time in the manuscript (DEGs and DEPs in line 21, K7 vs W7 in line 24, PHS in line 46). Please check through the manuscript and make adjustment for all abbreviations.
2. Citation is need for the statement in lines 135-137.
3. Caption for Figure 2 (line 158): (G) Heatmap analysis for DEGs in K0 vs. K7 group. Should it be K0 vs. W0?
4. Please stay with a consistent format when writing “P-value”. Inconsistent example: lines 162-163.
5. Correlation coefficient should be “r” instead of “R” (line 380).
6. Generally, the English expressions in this manuscript need be improved before published.
Author Response
Response to reviewer’ comment:
We would like to thank the guest editor and reviewers for the constructive comments that helped to significantly improve the enclosed revised version of our manuscript. We have reviewed and agreed with all the comments and suggestions made by the reviewers, and have addressed each comment separately. Additionally, English correction of the entire manuscript for grammar and sentence form has been completed by a native speaker. Please find below a detailed and itemized list of our responses to the points raised by the reviewers. Our responses to the reviewers’ questions/suggestions are highlighted in the revised manuscript in yellow.
Comments and Suggestions for Authors
Reviewer #2
Comments to the Author
This is a valuable study for wheat pre-harvest sprouting (PHS) resistance. Wheat PHS resistance is very important for wheat yield and end-use quality, especially in rain-fed areas. The authors performed RNA-sequencing and proteomic analysis for two wheat cultivars with very different sprouting rate at day 0 and day 7. DEGs and DEPs were identified from the transcriptomic and proteomic data, and gene ontology enrichment analysis for DEGs and DEPs elucidated pathways that contribute to PHS resistance. This study shed lights on biological pathways that associated with PHS resistance and provided candidate genes for future studies to understand the molecular mechanism for PHS resistance.
Major comments:
- The research used PHS susceptible lines “Keumgang” and PHS resistance line “Woori” to study their transcriptomic and proteomic differences at day 0 and day 7 of PHS inducement. The K7 sample was representative of sprouting process while the W7 sample was not (very few seeds germinated). Therefore, DEGs and DEPs identified in the K0 vs K7 comparison but not in the W0 vs W7 comparison should be more likely to be associated with sprouting. On the contrary, DEGs and DEPs identified in the W0 vs W7 comparison but not in the K0 vs K7 comparison should be more likely to be associated with sprouting resistance.
Answers: The authors agree with the reviewer’s comment. However, although W7 germinated poorly, we collected the germinated Woori samples and performed transcriptomic and proteomic analyses from a number of samples, as shown in Fig. 1D. Therefore, we expect that both K0 vs. K7 and W0 vs. W7 comparisons could be associated with sprouting. The DEG analyses in Fig. 3 and Fig. 4 also support our hypothesis, where the annotated DEGs were almost the same in K0 vs. K7 and W0 vs. W7. Therefore, we also maintain that the difference in sprouting tolerance between ‘Keumgang’ and ‘Woori’ should be explained through an observation of K7 vs. W7, rather than comparing K0 vs. K7 and W0 vs. W7.
- The 2.3 section in results is very long and descriptive, and all information can be found in Figure 3, which is a bit redundant. As a reader, I would like to see a summary of the GO enrichment analysis, i.e. what are the similarities across the DEGs in the four comparisons, what are the unique pathways in some comparisons, etc.
Answers: The authors agree with the reviewer's comments and have summarized Section 2.3. Gene Ontology results were mentioned as the main function of the manuscript, and an explanation was carried out over a larger range than before. [lines 181-226]
- The common genes between DEGs and DEPs seem to be very important. In the results, I suggest the authors at least provide a brief classification of those gene functions. Providing gene numbers is not very informative in the context.
Answers: The authors agree with the reviewer's points. A column was added for functional classification in Supplementary Table S6. Additionally, Supplementary Table S6 was rearranged to be Table 1 in the main text. In the renamed Table 1, the expression values for gene and protein are represented by red or green intensity according to expression level by two-fold change. [line 356]
- PHS resistance is affected not only by seed dormancy, but also by seed coat and spike structure. Genetic differences could not be identified in this study because the transcriptomic and proteomic data in this study could not capture the seed and spike development processes, which could be some limitations that need be mentioned in discussion.
Answers: The authors agree with the reviewer’s comment. The authors compared spike structure between two cultivars and added Supplementary Figure S5 for the morphologic comparison of spike structure. In addition, the authors also mentioned the overcoming of our limitation in the first paragraph of the Discussion section. [lines 379-395 and 715-716]
- It is very interesting that alternative splicing seems to play a role in responding to the abiotic stresses and germination. I wonder if it is possible to check if alternative splicing was observed in some of target genes (e.g. Vp-1) of the spliceosome proteins using the current transcriptomic data and proteomic data?
Answers: According to the reviewer’s suggestion, the authors investigated the alternative splicing of the PHS target genes Vp-1, DOG1, and MFT. In order to investigate the alternative splicing, authors conducted Integrative Genomics Viewer (IGV) analysis with read map raw data. Among the genes, MFT3B-1 and MFT3B-2 especially represented differential alternatively spliced transcripts (Supplementary Figure S4). The alternative splicing pattern of the two genes was similar in both species before PHS treatment (K0 and W0) in MFT-3B-1. However, after PHS treatment, the second and the third exons of MFT3B-1 were absent in K7 but present in W7, while the fourth exon was absent in W7 but present in K7. Additionally, the first exon in MFT3B-2 was lost in W7, which implies that there might be differences in how the spliceosome act in the both species. Authors also performed volcano plot and heatmap analysis of DEPs for alternative splicing in K7 vs W7 group (Supplementary Figure S4). A total of 42 alternative splicing unique variants were identified in W7. The authors added these results in the Results and Discussion sections. Authors have updated Supplementary Figure 1 with relative expression profiles and mixed qRT-PCR validation (Supplementary Figure 4). Therefore, Supplementary Figure 1 and 4 was rearranged to be Figure 8 in the main text. [lines 258-269, 510-530, 557 and 714-715]
Minor comments:
- Please provide full names for abbreviations as they are used first time in the manuscript (DEGs and DEPs in line 21, K7 vs W7 in line 24, PHS in line 46). Please check through the manuscript and make adjustment for all abbreviations.
Answers: The authors appreciate the reviewer’s kind comments. The authors added the full names for the first mentioned term. [lines 20-21 and 25-26]
- Citation is need for the statement in lines 135-137.
Answers: The authors deeply agree with the reviewer's comments. The proper references were added in the manuscript. [lines 137-139]
- Caption for Figure 2 (line 158): (G) Heatmap analysis for DEGs in K0 vs. K7 group. Should it be K0 vs. W0?
Answers: Yes, that is K0 vs. W0. The authors have replaced “K0 vs. K7” with “K0 vs. W0”. [line 169]
- Please stay with a consistent format when writing “P-value”. Inconsistent example: lines 162-163.
Answers: The authors have made a uniform and consistent format for “p-value”. [lines 178, 362, and 366]
- Correlation coefficient should be “r” instead of “R” (line 380).
Answers: The authors agree with the reviewer's point. The authors have changed 'r' to 'R'. [lines 349, 350, and 351]
- Generally, the English expressions in this manuscript need be improved before published.
Answers: The authors have asked an English editing service with a native speaker from the company MDPI to improve the English expression of the whole manuscript.

Reviewer 3 Report
The manuscript by Park et al. entitled “Transcriptome and Proteome Co-Profiling Offered Understanding of Pre-Harvest Sprouting (PHS) Molecular Mechanisms in Wheat (Triticum aestivum)” is providing an insight of the molecular mechanism of pre-harvest sprouting (PHS) which is in fact a physiological disorder and significantly reduce the wheat grain yield and quality. This manuscript is written well and such research scientifically sounds great and innovative by the way of incorporating both transcriptomic (RNAseq) and proteomic (LC-MS) data together to draw the significant concluding remarks by judging polyploid wheat genotypes having contrasting PHS sensitivity. This article is not without its minor drawbacks, which I am describing below:
Figure 3: Please increase the resolution. It is very hard to see the words. Or some graphs can be transferred to supplementary by keeping the major ones in the main figure.
Figure 5: You can place 'E,F,G, H' under 'A,B,C,D' to make the whole figure a portrait form, not in landscape.
Figure 6: Please reduce the space between the graphs to make it more compact and easy to visualise the words.
Line 559: Please mention which type of cultivar is it? Durum or bread wheat?
Line 566: How much relative humidity (RH%) was maintained?
Line 684: Please correct the '2' as non-superscript. It should be like '2-ΔΔCt'.
After correcting these minor changes, this manuscript can be publishable in Plants.
Author Response
Response to reviewer’ comment:
We would like to thank the guest editor and reviewers for the constructive comments that helped to significantly improve the enclosed revised version of our manuscript. We have reviewed and agreed with all the comments and suggestions made by the reviewers, and have addressed each comment separately. Additionally, English correction of the entire manuscript for grammar and sentence form has been completed by a native speaker. Please find below a detailed and itemized list of our responses to the points raised by the reviewers. Our responses to the reviewers’ questions/suggestions are highlighted in the revised manuscript in yellow.
Comments and Suggestions for Authors
Reviewer #3
Comments to the Author
The manuscript by Park et al. entitled “Transcriptome and Proteome Co-Profiling Offered Understanding of Pre-Harvest Sprouting (PHS) Molecular Mechanisms in Wheat (Triticum aestivum)” is providing an insight of the molecular mechanism of pre-harvest sprouting (PHS) which is in fact a physiological disorder and significantly reduce the wheat grain yield and quality. This manuscript is written well and such research scientifically sounds great and innovative by the way of incorporating both transcriptomic (RNAseq) and proteomic (LC-MS) data together to draw the significant concluding remarks by judging polyploid wheat genotypes having contrasting PHS sensitivity. This article is not without its minor drawbacks, which I am describing below:
Major comments:
- Figure 3: Please increase the resolution. It is very hard to see the words. Or some graphs can be transferred to supplementary by keeping the major ones in the main figure.
Answers: We apologize for your inconvenience. The authors have replaced all figures, including Figure 3, with high-definition versions, and have increased the font size. Additionally, three Supplementary Figures have been added into the manuscript. [line 228]
- Figure 5: You can place 'E,F,G,H' under 'A,B,C,D' to make the whole figure a portrait form, not in landscape.
Answers: The authors have arranged Figure 5 in a vertical form, as requested by the reviewer. [line 290]
- Figure 6: Please reduce the space between the graphs to make it more compact and easy to visualise the words.
Answers: The authors agree with the reviewer's point. Figure 6 has been placed with a smaller spacing than in the original manuscript, and the font size has been increased. [line 333]
- Line 559: Please mention which type of cultivar is it? Durum or bread wheat?
Answers: The Korean wheat varieties used in this study were common wheat. Therefore, the authors have included 'common wheat' in the manuscripts to provide readers with the exact information. [line 115]
- Line 566: How much relative humidity (RH%) was maintained?
Answers: The authors have confirmed the relative humidity for the experimental conditions. The humidity conditions in the experiment were set to 99% humidity using an artificial rainfall system in the greenhouse, and the actual relative humidity was about 90%. [lines 132-134]
- Line 684: Please correct the '2' as non-superscript. It should be like '2-ΔΔCt'.
Answers: The authors have changed the superscript formatting to '2-ΔΔCt'. [line 691]

Round 2
Reviewer 2 Report
The authors did a lot of work and editing to improve the manuscript. The results for the GO enrichment analysis have been re-organized and presented more concisely. The overlapping between DEGs and DEPs have been emphasized and highlighted in the main table, which conveys the take-home message in an efficient way. I really appreciate and am excited about the discovery of the alternative splicing patterns of MFT-3B-1 and MFT-3B-2, which validated the hypothesis raised from the GO enrichment analysis and provided insights into the regulatory mechanism of important genes for PHS resistance.
To improve Supplemental Figure S4, I suggest the authors add gene models (the representative gene models) for MFT-3B-1 and MFT-3B-2 so that it is clear how the alternative splicing happens.
A minor place to change: the correlation coefficient (r) need be italicized.
Author Response
We like to thank the editor and reviewers for the constructive comments which helped to significantly improve the enclosed revised version. Authors have reviewed the comments and suggestions made by the reviewer have addressed each comment. Please find itemized list of our responses to the points raised by the editor and the reviewer. Our responses to the reviewers’ questions/suggestions are highlighted in the revised manuscript in green.
Comments to the Author
This is a valuable study for wheat pre-harvest sprouting (PHS) resistance. Wheat PHS resistance is very important for wheat yield and end-use quality, especially in rain-fed areas. The authors performed RNA-sequencing and proteomic analysis for two wheat cultivars with very different sprouting rate at day 0 and day 7. DEGs and DEPs were identified from the transcriptomic and proteomic data, and gene ontology enrichment analysis for DEGs and DEPs elucidated pathways that contribute to PHS resistance. This study shed lights on biological pathways that associated with PHS resistance and provided candidate genes for future studies to understand the molecular mechanism for PHS resistance.
Answers: The authors would like to thank the reviewer once again for their detailed and accurate review.
Major comments:
- To improve Supplemental Figure S4, I suggest the authors add gene models (the representative gene models) for MFT-3B-1 and MFT-3B-2 so that it is clear how the alternative splicing happens.
Answers: The author agrees with the reviewer's suggestion. Alternative splicing-related gene models of MFT-3B-1 and MFT-3B-2 have been added to Supplementary Figure 4 and figure legend [lines714-715].
- A minor place to change: the correlation coefficient (r) need be italicized.
Answers: The authors agree with the reviewer's point. The authors have changed 'r' to 'r'. [lines 349, 350, and 351]
